# GENERALIZED POLICY ITERATION USING TENSOR APPROXIMATION FOR HYBRID CONTROL

**Suhan Shetty**[1,2], **Teng Xue**[1,2], **Sylvain Calinon**[1,2]

[1] Idiap Research Institute, Martigny, Switzerland

[2] École Polytechnique Fédérale de Lausanne (EPFL), Switzerland

`name.surname@idiap.ch`

## ABSTRACT

Optimal Control of dynamic systems involving hybrid actions is a challenging task in robotics. To address this, we present a novel algorithm called Generalized Policy Iteration using Tensor Train (TTPI) that belongs to the class of Approximate Dynamic Programming (ADP). We use a low-rank tensor approximation technique called Tensor Train (TT) to approximate the state-value and advantage function which enables us to efficiently handle hybrid action space. We demonstrate the superiority of our approach over previous baselines for some benchmark problems with hybrid action spaces. Additionally, the robustness and generalization of the policy for hybrid systems are showcased through a real-world robotics experiment involving a non-prehensile manipulation task.

Robotic systems often exhibit complex nonlinear dynamics that may involve hybrid actions. The need for real-time control, high precision, and adequate robustness to cope with disturbances or changes in the environment can result in demanding computational requirements that are challenging to meet with classical control methods. Optimal Control (OC) based on the principles of Dynamic Programming (DP) is a popular tool in robotics but they are still limited to systems with continuous actions and differentiable dynamics.

Approximate DP (ADP) and Reinforcement Learning (RL) overcome the curse of dimensionality faced by classical DP algorithms by using function approximation techniques (Sutton & Barto, 2005; Bertsekas, 2012). OC is closely related to ADP and uses the system's model to obtain an optimal policy, while RL focuses on learning a policy through trial-and-error interactions with the environment. Both methods aim to find a compact representation of the value functions to obtain a control policy. ADP faces difficulty in approximating the value function throughout the entire state space, conversely, RL restricts its approximation to a smaller region where data is collected, resulting in limited generalizability but greater scalability. However, the existing approaches for both ADP and RL face challenges in handling hybrid action space. Furthermore, existing ADP approaches also find it challenging to cope with large action spaces and hybrid states.

In this paper, we present a novel ADP algorithm, called Generalized Policy Iteration using Tensor Train (TTPI) which overcomes the challenges faced by existing ADP methods for hybrid system control. TTPI is an approximate version of the Generalized Policy Iteration (GPI) algorithm—a DP algorithm that encompasses both Value Iteration (VI) and Policy Iteration (PI) algorithms. We use Tensor Train (TT) (Oseledets, 2011), a low-rank tensor approximation technique (Grasedyck et al., 2013), to model the state-value and the advantage function.

TT is a versatile function approximator that allows us to simultaneously handle continuous and discrete state and action variables. It approximates a given function as a sum of products of univariate functions, allowing for fast algebraic operations and interpretation. The use of TT-Cross (Oseledets & Tyrtyshnikov, 2010; Savostyanov & Oseledets, 2011), a powerful gradient-free method to approximate functions in TT format in a nonparametric manner, allows us to achieve TT approximation of state-value and advantage function with a desired accuracy in a fast manner, thus exploiting the knowledge of the system model and the reward function. Moreover, the TT representation of the advantage function enables us to use optimization techniques such as TTGO (Shetty et al., 2023) to retrieve policies for hybrid action spaces.

The TT representation is particularly effective when the function being approximated is smooth, resulting in a low-rank representation in the TT format. Our experiments demonstrate that such property is frequently observed in ADP while dealing with hybrid systems. Indeed, even though the system dynamics and reward functions may be non-smooth and discontinuous, the optimal value functions typically exhibit low-rank structures.

**Contributions:** We introduce TTPI, a novel ADP algorithm for optimal control that leverages TT as a function approximator to address the challenges of hybrid system control in robotics. Our approach is interpretable and eliminates the need for differentiability of the system dynamics and reward function which is a common assumption in the existing ADP algorithms. Our experiments demonstrate that TTPI outperforms state-of-the-art algorithms in terms of both training time and performance on various benchmark control tasks for hybrid control. To showcase the practicality and generalization of our approach, we conducted a real-world robotic experiment where we successfully tackled a non-prehensile planar manipulation task that is notoriously difficult for existing control methods. Our results demonstrate the robustness of the policy and highlight the potential of our approach to addressing complex control problems in robotics.

# 1 GENERALIZED POLICY ITERATION USING ADVANTAGE FUNCTION

## 1.1 THE OPTIMAL CONTROL PROBLEM

We consider a discrete-time dynamic system with $d$-dimensional state space and $m$-dimensional action space. For ease of presentation, we assume the dynamic system to be deterministic, however, our approach can also handle a stochastic model (see Section 2.9).

We denote the state at time $t$ by $\boldsymbol{s}_t = (s_t^1, \ldots, s_t^d)$, and action by $\boldsymbol{a}_t = (a_t^1, \ldots, a_t^m)$. The dynamics of the system is given by

$$
\begin{aligned}
\boldsymbol{s}_{t+1} &= f(\boldsymbol{s}_t, \boldsymbol{a}_t), \\
\text{s.t.} \quad s_t^i &\in \Omega_{s^i}, \forall i \in \{1, \ldots, d\}, \\
a_t^j &\in \Omega_{a^j}, \forall j \in \{1, \ldots, m\},
\end{aligned}
\tag{1}
$$

where the domain of each state $\Omega_{s^i}$ and action $\Omega_{a^i}$ can be a bounded real interval or a discrete set. Let $\Omega_{\boldsymbol{s}}$ denote the state space and $\Omega_{\boldsymbol{a}}$ denote the action-space.

Let $r(\boldsymbol{s}, \boldsymbol{a})$ represent the reward function and $\Delta t$ be the time step for the discrete-time control. We define $R(\boldsymbol{s}, \boldsymbol{a}) = r(\boldsymbol{s}, \boldsymbol{a})\Delta t$. Our goal is to obtain an optimal policy $\pi^*$ for the following infinite horizon optimal control problem for any given initial state in the state space $\boldsymbol{s}_0 \in \Omega_{\boldsymbol{s}}$:

$$
\begin{aligned}
\pi^* &= \arg\max_{\pi} \sum_{t=0}^{\infty} \gamma^t R(\boldsymbol{s}_t, \pi(\boldsymbol{s}_t)), \ \forall \boldsymbol{s}_0, \\
\text{with} \quad \boldsymbol{s}_{t+1} &= f(\boldsymbol{s}_t, \pi(\boldsymbol{s}_t)),
\end{aligned}
\tag{2}
$$

where $\gamma$ is the discount factor $0 \le \gamma < 1$.

We do not make any assumption on the structure or differentiability of the dynamics $f$ and the reward function $r$. For example, a black box deterministic simulator that returns the next state and the reward for the state-action pair satisfies our requirement. However, for a fast implementation of our algorithm described in Section 2.8, the simulator should ideally process a batch of state-action pairs for parallel implementation.

## 1.2 DYNAMIC PROGRAMMING

The state-value function $V^\pi$ corresponding to a policy $\pi$, with discount factor $\gamma$, is defined as

$$
\begin{aligned}
V^\pi(\boldsymbol{s}_0) &= \sum_{t=0}^{\infty} \gamma^t R(\boldsymbol{s}_t, \pi(\boldsymbol{s}_t)), \ \forall \boldsymbol{s}_0, \\
\text{where} \quad \boldsymbol{s}_{t+1} &= f(\boldsymbol{s}_t, \pi(\boldsymbol{s}_t)), \ \forall t.
\end{aligned}
\tag{3}
$$

Given a state-value function $V : \Omega_s \to \mathbb{R}$, a policy $\pi$ and the discount factor $\gamma$, the Bellman operator $\mathcal{B}^\pi$ is a functional that is defined as $\mathcal{B}^\pi V(\boldsymbol{s}) = R(\boldsymbol{s}, \pi(\boldsymbol{s})) + \gamma V(f(\boldsymbol{s}, \pi(\boldsymbol{s}))), \forall \boldsymbol{s} \in \Omega_{\boldsymbol{s}}$ where $\mathcal{B}^\pi : V \to V$.

We define the advantage function $A_V$ corresponding to the value function $V$ as

$$A_V(\boldsymbol{s}, \boldsymbol{a}) = R(\boldsymbol{s}, \boldsymbol{a}) + \gamma\Big(V\big(f(\boldsymbol{s}, \boldsymbol{a})\big) - V(\boldsymbol{s})\Big), \ \forall(\boldsymbol{s}, \boldsymbol{a}) \in \Omega_{\boldsymbol{s}} \times \Omega_{\boldsymbol{a}}. \qquad (4)$$

### 1.3 Challenges in Approximate Dynamic Programming

Algorithm 1 describes the value iteration (VI) algorithm (Sutton & Barto, 2005), a popular DP algorithm.

One of the challenges in implementing the VI algorithm and other similar DP algorithms including the Policy Iteration (PI) algorithm (Sutton & Barto, 2005) in practice is the curse of dimensionality in representing the value function when the involved state space is either high-dimensional or includes continuous states. ADP addresses this challenge by using function approximation techniques.

In addition, retrieving the policy $\pi^k$ from the advantage function is difficult if it is nonconvex, if there are bounds on the actions, if the action space is large, or if the action space is hybrid. An inefficient optimization technique for policy retrieval increases the overall time of the algorithm, as it must be repeated for each state in every iteration, and it results in a sub-optimal policy. The lack of such policy retrieval techniques is a bottleneck in the development of ADP algorithms for hybrid control.

---

**Algorithm 1** VI Algorithm

---

**Input:** Initial value function $V^0$, convergence threshold $\epsilon$
**Output:** Optimal policy $\pi^*$
1: Set $k = 0$
2: **repeat**
3:     $\pi^k(\boldsymbol{s}) := \arg\max_{\boldsymbol{a}} A_{V^k}(\boldsymbol{s}, \boldsymbol{a})$
4:     $V^{k+1} = \mathcal{B}^{\pi_k} V^k$
5:     **if** $\|V^{k+1} - V^k\|_\infty < \epsilon$ **then**
6:         **break**
7:     **end if**
8:     Set $k \leftarrow k + 1$
9: **until** convergence
10: $V^* = V^k$
11: $\pi^*(\boldsymbol{s}) = \arg\max_{\boldsymbol{a}} A_{V^*}(\boldsymbol{s}, \boldsymbol{a})$

---

## 2 Generalized Policy Iteration using Tensor Train (TTPI)

In this section, we briefly describe the proposed approach and the related concepts used to tackle the previously described challenges in ADP for handling hybrid actions and large action spaces. Further details are provided in the Appendix. In summary, we overcome the challenges mentioned in the ADP algorithms using TT as a function approximator. We propose to model the advantage function explicitly in TT format and use TTGO, a technique for optimization of functions in TT format proposed by Shetty et al. (2023); Chertkov et al. (2022), for policy retrieval.

### 2.1 Tensors as Discrete Analogue of a Function

A multivariate function $P(x_1, \ldots, x_d)$ defined over a rectangular domain made up of the Cartesian product of intervals (or discrete sets) $I_1 \times \cdots \times I_d$ can be discretized by evaluating it at points in the set $\mathcal{X} = \{(x_1^{i_1}, \ldots, x_d^{i_d}) : x_k^{i_k} \in I_k, i_k \in \{1, \ldots, n_k\}\}$. This gives us a tensor $\mathcal{P}$, a discrete version of $P$, where $\mathcal{P}_{(i_1, \ldots, i_d)} = P(x_1^{i_1}, \ldots, x_d^{i_d}), \forall(i_1, \ldots, i_d) \in \mathcal{I}_{\mathcal{X}}$, and $\mathcal{I}_{\mathcal{X}} = \{(i_1, \ldots, i_d) : i_k \in \{1, \ldots, n_k\}, k \in \{1, \ldots, d\}\}$. The value of $P$ at any point in the domain can then be approximated by interpolating between the elements of the tensor $\mathcal{P}$.

### 2.2 Tensor Decomposition

Representing a high-dimensional tensor is difficult because of the limitation in storage. Tensor decomposition techniques (Kolda & Bader, 2009; Sidiropoulos et al., 2017) solve this problem by representing the tensor using a smaller number of lower-dimensional tensors, known as factors, which occupy less memory. These factors are combined with certain algebraic operations, depending on the decomposition method, to represent the elements of the original tensor. In addition to the compact representation, they also enable efficient algebraic operations in the compressed format.

The accuracy of a tensor representation is usually controlled by its rank in the decomposition, which is proportional to the number of elements in the factorization. The rank of the tensor is closely related to the separability of the underlying function. In practice, for continuous variables, the smoothness of the underlying function often corresponds to a low rank.

## 2.3 TT Decomposition

TT decomposition, also known as Matrix Product State (MPS) or Tensor Networks (TN) in physics (Cichocki et al., 2016), is a widely used tensor decomposition technique due to its versatility and effective methods for determining the approximation.

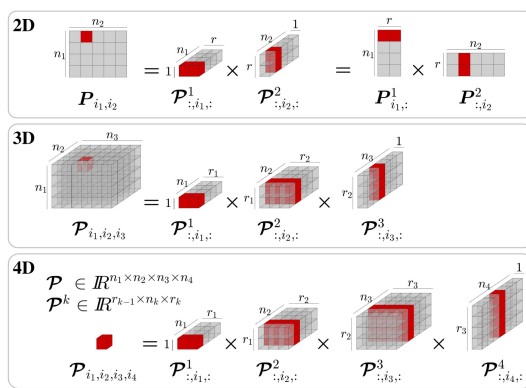

Figure 1: TT decomposition extends matrix decomposition techniques to arrays with higher dimensions. In matrix decomposition, an element in the original matrix can be obtained by multiplying the appropriate rows or columns of the factors. Likewise, an element in a tensor represented in TT format can be retrieved by multiplying selected slices of the core tensors, which are the factors. The illustration shows examples of 2nd, 3rd, and 4th-order tensors.

TT decomposition approximates a given tensor (a multidimensional array) compactly using a set of third-order tensors called *cores*. A $d$-th order tensor $\mathcal{P} \in \mathbb{R}^{n_1 \times \cdots \times n_d}$ in TT format is represented using a tuple of $d$ third-order tensors $(\mathcal{P}^1, \ldots, \mathcal{P}^d)$. The dimension of the cores are given as $\mathcal{P}^1 \in \mathbb{R}^{1 \times n_1 \times r_1}, \mathcal{P}^k \in \mathbb{R}^{r_{k-1} \times n_k \times r_k}, k \in \{2, \ldots, d-1\}$, and $\mathcal{P}^d \in \mathbb{R}^{r_{d-1} \times n_d \times 1}$ with $r_0 = r_d = 1$. As shown in Figure 1, we can access the element $(i_1, \ldots, i_d)$ of the tensor in this format simply given by multiplying matrix slices from the cores

$$\mathcal{P}_{(i_1,\ldots,i_d)} = \mathcal{P}^1_{:,i_1,:} \mathcal{P}^2_{:,i_2,:} \cdots \mathcal{P}^d_{:,i_d,:}, \tag{5}$$

where $\mathcal{P}^k_{:,i_k,:} \in \mathbb{R}^{r_{k-1} \times r_k}$ represents the $i_k$-th frontal slice (a matrix) of the third-order tensor $\mathcal{P}^k$. The dimensions of the cores are such that the above matrix multiplication yields a scalar. The *TT-rank* of the tensor in TT representation is then defined as the tuple $r = (r_1, r_2, \ldots, r_{d-1})$. We call $r = \max(r_1, \ldots, r_{d-1})$ as the *maximal rank*. For any given tensor, there always exists a TT decomposition (5) (Oseledets, 2011).

## 2.4 TT-Cross

TT-Cross (Oseledets & Tyrtyshnikov, 2010; Savostyanov & Oseledets, 2011) efficiently computes a TT approximation of a tensor with controlled accuracy by evaluating only a small number of its elements, without requiring the entire tensor to be stored in memory. It does this by computing only specific tensor fibers at a time, in a black-box manner. A noteworthy feature of TT-Cross is that it is an unsupervised and nonparametric approach as it directly takes the function being modeled as its input and the number of parameters in its TT representation is adjusted based on the structure of the underlying function until a specified accuracy of the approximation is obtained.

Consider a tensor $\mathcal{P}$ that is a discrete analog of a function $P$ with domain $\Omega$, using a discretization set $\hat{\Omega}$. Instead of evaluating the entire tensor, TT-Cross selects a subset of elements by evaluating function $P$ at various points in the discretization set of $\hat{\Omega}$. The approximate tensor in TT format, $\hat{\mathcal{P}} = \text{TT-cross}(P, \hat{\Omega}, r_{\max}, \epsilon,)$, requires only $\mathcal{O}(ndr^2)$ evaluations, rather than $\mathcal{O}(n^d)$ evaluations of the full tensor. Here, $r_{\max}$ is the upper-bound on the rank of the TT, $\epsilon$ is the accuracy of the approximation, and $r$ is the actual rank of the tensor in TT format. TT-Cross is efficient when the TT-rank $r$ is low, which is often the case for state-value functions in many problems involving hybrid states. We can then obtain the continuous approximation to $P$ from $\hat{\mathcal{P}}$ using (7).

## 2.5 Algebraic operations in TT format

In addition to representing a tensor compactly, the TT format enables the efficient execution of various tensor algebraic operations in its compact form. Operations such as addition, subtraction, and inner product of TT tensors can be performed efficiently (Lee & Cichocki, 2018). The mean, norm, and gradient can also be calculated efficiently. The compactness of the representation can be improved by using *TT-rounding* operations, with a trade-off in accuracy. The availability of these

algebraic tools allows a thorough analysis of the functions represented in TT format, thus making them highly interpretable.

## 2.6 CONTINUOUS APPROXIMATION USING TT

Given the discrete analogue tensor $\mathcal{P}$ of a function $P$, we obtain the continuous approximation by spline-based interpolation of the TT cores corresponding to the continuous variables only. For example, we can use linear interpolation for the cores (i.e., between the matrix slices of the core) and define a matrix-valued function corresponding to each core $k \in \{1, \ldots, d\}$,

$$\boldsymbol{P}^k(x_k) = \frac{x_k - x_k^{i_k}}{x_k^{i_k+1} - x_k^{i_k}} \boldsymbol{\mathcal{P}}^k_{:,i_k+1,:} + \frac{x_k^{i_k+1} - x_k}{x_k^{i_k+1} - x_k^{i_k}} \boldsymbol{\mathcal{P}}^k_{:,i_k,:}, \tag{6}$$

where $x_k^{i_k} \leq x_k \leq x_k^{i_k+1}$ and $\boldsymbol{P}^k : I_k \subset \mathbb{R} \to \mathbb{R}^{r_{k-1} \times r_k}$ with $r_0 = r_d = 1$. This induces a continuous approximation of $P$ given by

$$P(x_1, \ldots, x_d) \approx \boldsymbol{P}^1(x_1) \cdots \boldsymbol{P}^d(x_d). \tag{7}$$

This allows us to selectively do the interpolation only for the cores corresponding to continuous variables, and hence we can represent functions in TT format whose variables could be a mix of continuous and discrete elements.

We overload the terminology to define the continuous TT representation as

$$\begin{aligned} \boldsymbol{\mathcal{P}}^k_{:,\,x_k,\,:} &= \boldsymbol{P}^k(x_k), \\ \boldsymbol{\mathcal{P}}_{\boldsymbol{x}} &= P(x_1, \ldots, x_d), \text{ where } \boldsymbol{x} = (x_1, \ldots, x_d), \\ &= \boldsymbol{\mathcal{P}}^1_{:,\,x_1,\,:} \cdots \boldsymbol{\mathcal{P}}^d_{:,\,x_d,\,:}, \quad \forall x_k \in \Omega_{x_k}, \, \forall k \in \{1, \ldots, d\}. \end{aligned} \tag{8}$$

## 2.7 TTGO: OPTIMIZATION OF FUNCTION IN TT-FORMAT

In addition to the availability of algorithms like TT-Cross for finding function approximation and the accompanying algebraic tools, an advantage of using TT decomposition for approximating functions in ADP is its ability to efficiently find optima over a mix of continuous and discrete variables. This was introduced as Tensor Train for Global Optimization (TTGO) in Shetty et al. (2023), in the form of a stochastic method. A deterministic version of this was proposed in Chertkov et al. (2022). We propose improvements to this approach in this paper, see Section A.9 in Appendix. In practice, the technique often yields globally optimal solutions, as demonstrated in Shetty et al. (2023) and Chertkov et al. (2022).

The basic idea behind TTGO is that it transforms the given function in TT format, using the accompanying necessary algebraic tools, into a nonnegative function in TT format that can be interpreted as a probability density function. The efficient sampling techniques for density functions in TT format allow us to pick samples of only high-density regions which in turn correspond to the optima. In practice, the chosen number of prioritized samples $N \geq 1$ and the sample(s) with the highest density (or least cost) is used to represent the optima. The solution obtained from such a procedure can be refined further using local optimization techniques such as Newton-type optimization for continuous variables. But, in practice, as in this paper, the fine-tuning is often not required.

In this paper, we identify and exploit TTGO's ability to handle a mix of continuous and discrete variables. In addition, we perform optimization in the batch form: we propose to model the advantage function $A(\boldsymbol{s}, \boldsymbol{a})$ in ADP in TT format, and adapt TTGO to obtain the optimal actions $\boldsymbol{a}$ corresponding to a batch of states $\boldsymbol{s}$ (i.e. parallel computation of $\arg\max_{\boldsymbol{a}} A(\boldsymbol{s}, \boldsymbol{a})$ ) in an efficient manner.

## 2.8 TTPI ALGORITHM

By combining the conceptual ideas proposed so far, Algorithm 2 presents the TTPI algorithm, which addresses the previously mentioned challenges in ADP using TT as the function approximator for state-value and advantage functions and TTGO for policy retrieval.

In the TTPI algorithm, the value update step involves computing $\pi^k(\boldsymbol{s})$ (i.e., $\arg\max_{\boldsymbol{a}} A_{V^k}(\boldsymbol{s}, \boldsymbol{a})$) numerous times across several iterations. To compute $V_j^k$ in TT-format, the function $\mathcal{B}^{\pi_k} V_{j-1}^k$ is queried iteratively using TT-Cross($\mathcal{B}^{\pi_k} V_{j-1}^k, r_{\max}, \epsilon$), with batches of states (usually ranging from 10,000 to 100,000 in practice). This requires computing the policy $\pi^k$ for each of these states in batch form. We use TT-round to compress the value functions in TT format at the end of every policy

evaluation (i.e., after updating the value function for the current policy). We use cubic spline-based interpolation for continuous variables which reduces the number of discretization points required by TT-cross to construct the TT model.

To resolve the bottleneck in policy retrieval, we propose to compute the advantage function $A_{V^k}$ in TT format using TT-Cross. This is efficient as the calculation only requires evaluating $V^k$ and $R(\boldsymbol{s}, \boldsymbol{a})$, which are cheap to compute. This enables the numerical optimization of variables for functions in TT format using TTGO, as outlined in Section 2.7. As a result, $\pi^k(\boldsymbol{s})$ over batches of states can be obtained quickly. Most importantly, this allows us to handle hybrid action space. The computational cost involved in retrieving a solution is $\mathcal{O}(Nmdr_{\max}^2)$ which is linear in the number of discretizations ($d$) of an action variable and the dimension of action space ($m$).

The computational cost of the algorithm increases linearly with the number of dimensions in both state and action spaces and grows quadratically with the rank of the functions represented in TT format, thanks to the properties of TT-Cross and TT-representation. A PyTorch-based GPU-accelerated implementation of these algorithms is provided along with the supplementary material at https://sites.google.com/view/ttpi4control.

### 2.9 TTPI for Stochastic Systems

In this section, we show how our approach can be extended to consider stochastic system dynamics. Instead of relying on deterministic system dynamics of the form $\boldsymbol{s}' = f(\boldsymbol{s}, \boldsymbol{a})$, we consider the transition probability $P(\boldsymbol{s}', \boldsymbol{s}, \boldsymbol{a})$ and the reward function $R(\boldsymbol{s}, \boldsymbol{a})$ in TT format. The transition probability $P(\boldsymbol{s}', \boldsymbol{s}, \boldsymbol{a})$ can be obtained by fitting a density model to data collected from the robot. To achieve this, we can employ the TT format for density modeling as suggested by Novikov et al. (2021) and Han et al. (2018). Alternatively, if the function $P$ is available in a different format such as NN, we can utilize TT-Cross. By leveraging the algebraic tools provided in TT format, we can normalize $P$ such that $\sum_{\boldsymbol{s}'} P(\boldsymbol{s}', \boldsymbol{s}, \boldsymbol{a}) = 1$ (or integrate if $\boldsymbol{s}'$ is continuous). The following outlines the procedure to update the value function and policy under this approach:

$$
\begin{aligned}
V^k =& \text{TT-Cross}(U^k, \hat{\Omega}_{\boldsymbol{s}}, r_{\max}, \epsilon), \\
U^k(\boldsymbol{s}) =& R(\boldsymbol{s}, \pi^k(\boldsymbol{s})) + \gamma W^k(\boldsymbol{s}, \pi^k(\boldsymbol{s})), \\
W^k(\boldsymbol{s}, \boldsymbol{a}) =& \sum_{\boldsymbol{s}'} P(\boldsymbol{s}', \boldsymbol{s}, \boldsymbol{a}) V^k(\boldsymbol{s}'), \\
A_{V^k}(\boldsymbol{s}, \boldsymbol{a}) =& R(\boldsymbol{s}, \boldsymbol{a}) + \gamma(W^k(\boldsymbol{s}, \boldsymbol{a}) - V^k(\boldsymbol{s})), \\
\pi^k(\boldsymbol{s}) =& \arg\max_{\boldsymbol{a}} A_{V^k}(\boldsymbol{s}, \boldsymbol{a}).
\end{aligned}
\tag{9}
$$

In the above algorithm, as $P$ and $V^k$ are both in TT format, we can obtain $W^k$ efficiently by using algebraic operation over TT format (namely, element-wise product and contraction operations over $\boldsymbol{s}'$). Then $A_{V^k}$ can be readily computed in TT-format using addition operations over the TT tensors as $R$, $W^k$, and $V^k$ are also in TT format. We only need TT-cross to find $V^k$. Hence the algorithm would be very efficient if $P$ is known in TT format. However, we acknowledge that, in practice, obtaining a stochastic model of a system $P$ from data is a challenging problem and it is still an ongoing area of research.

## 3 Experiments

In our experiments, we utilized an NVIDIA GeForce RTX 3090 GPU with 24GB of memory. For the applications considered, we discretized each continuous variable with 100 points using uniform discretization. To approximate the value and advantage functions in TT format using TT-Cross, an accuracy of $\epsilon = 10^{-3}$ proved sufficient. We set $r_{\max}$ to a large value of 100. The discount factor was chosen in the range of 0.99 to 0.9999, depending on the time step $\Delta t$ which ranged from 0.01 to 0.001. The rank of the value function in the applications considered ranged between 5 to 50, and the rank of the advantage function was roughly twice that of the value function.

### 3.1 Simulation Experiments:

**Baseline:** To the best of our knowledge, there are no established approaches for OC based on ADP algorithms that can handle hybrid actions. To evaluate our algorithm performance, we compared it against Deep RL techniques for hybrid action spaces such as HyAR, HPPO and PDQN (Li et al.,

---

**Algorithm 2** TTPI: Generalized Policy Iteration using Tensor Train

**Input:**
  $n_v$: Number of value update steps
  $\epsilon$: Accuracy of TT representation
  $r_{\max}$: Maximum TT-rank
  $\delta_{\max}$: Convergence tolerance
  $r(\boldsymbol{s}, \boldsymbol{a})$: Reward function
  $\Delta t$: Time Discretization
  $f(\boldsymbol{s}, \boldsymbol{a})$: Forward simulation
  $\hat{\Omega}_{\boldsymbol{s}}$: Discretization of state space
  $\hat{\Omega}$: Discretization of state-action space
  $(\hat{\Omega}_{\boldsymbol{s}} \times \hat{\Omega}_{\boldsymbol{a}})$
  $N$: Number of candidate samples for optima used in TTGO.

**Initialize:**
1: Initialize $V^0 = 0$ in TT-format
2: Initialize Advantage model:
3: $A_{V^0} = \text{TT-Cross}(R(\boldsymbol{s}, \boldsymbol{a}), \hat{\Omega}, r_{\max}, \epsilon)$
4: (alternatively, initialize arbitrarily),
5: Set $k = 0$

**Output:** Policy $\pi^*$
1: **while** $\delta \leq \delta_{\max}$ **do**
2: $\quad k = k + 1$
3: $\quad \pi^k(\boldsymbol{s}) := \underset{\boldsymbol{a}}{\arg\max}\, A_{V^{k-1}}(\boldsymbol{s}, \boldsymbol{a})$ (Use TTGO)
4: $\quad V_0^k = V^{k-1}$
5: $\quad$ **for** $j \leftarrow 1$ to $n_v$ **do**
6: $\quad\quad V_j^k(\boldsymbol{s}) = \text{TT-Cross}(\mathcal{B}^{\pi^k} V_{j-1}^k, \hat{\Omega}_{\boldsymbol{s}}, r_{\max}, \epsilon)$
7: $\quad$ **end for**
8: $\quad V^k = \text{TT-round}(V_{n_v}^k, \epsilon)$
9: $\quad A^k(\boldsymbol{s}, \boldsymbol{a}) = R(\boldsymbol{s}, \boldsymbol{a}) + \gamma\Big(V^k\big(f(\boldsymbol{s}, \boldsymbol{a})\big) - V^k(\boldsymbol{s})\Big)$
10: $\quad A_{V^k} = \text{TT-Cross}(A^k, \hat{\Omega}, r_{\max}, \epsilon)$
11: $\quad \delta = \frac{\|V^k - V^{k-1}\|_2}{\|V^{k-1}\|_2}$
12: **end while**
13: Set $V^* = V^k$
14: $\pi^*(\boldsymbol{s}) = \underset{\boldsymbol{a}}{\arg\max}\, A_{V^*}(\boldsymbol{s}, \boldsymbol{a})$

---

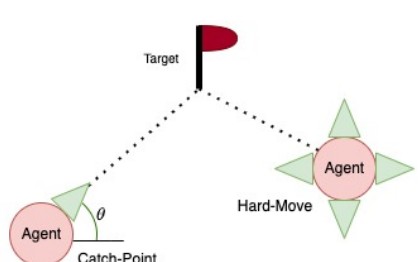

Figure 2: The tasks considered in this study involve controlling an agent to reach a target point in a 2D space. In the first task, called "Catch-Point", the agent has control over its heading direction (continuous) and the option to either stop or move toward the target (binary variable). In the second task, known as "Hard-Move", the agent is equipped with $n$ actuators, and it can decide to activate or deactivate each actuator ($n$ binary actions) and specify acceleration along each actuator ($n$ continuous variables).

2022; Fan et al., 2019; Xiong et al., 2018). The HyAR algorithm has shown superiority over other Deep RL techniques for high-dimensional hybrid action spaces. It is important to acknowledge that TTPI assumes access to the system dynamics and the reward function, whereas Deep RL techniques, in theory, are agnostic to the system model and implicitly address a more challenging problem than TTPI. However, many of these methods are data-inefficient and, like TTPI, assume access to a simulator.

**Evaluation:** We evaluated our algorithm on two benchmark problems involving systems with hybrid action spaces: the Catch-Point (CP) Problem and the Hard-Move (HM) problem, as proposed by Li et al. (2022). The Catch-Point Problem has four states and an action space with one discrete and one continuous action. The Hard-Move problem has $n$ actuators, resulting in a total of $2n$ action variables, with $n$ binary and $n$ continuous variables. Thus, this problem allows testing the scalability for high-dimensional action spaces by increasing $n$.

The results, as presented in Table 1, provide strong evidence of TTPI's superior performance compared to the baseline method. TTPI demonstrates faster training times and generates highly performant policies. In contrast, the baseline method struggles with generalization and produces lower-quality solutions, particularly for the Hard-Move problem with a number of actuators $m > 12$. This is attributed to TT-Cross accurately modeling the value functions by leveraging the system model and reward function, in a fast manner and efficient policy retrieval using TTGO.

## 3.2 REAL ROBOT EXPERIMENTS
We demonstrate the effectiveness of our proposed method for hybrid system control on a planar pushing task with a face-switching mechanism (Xue et al., 2023) and involves discrete states and actions. The objective is to push a block with freedom in switching both the contact modes and faces.

|  | $d$ | $m$ | HPPO | | | PDQN | | | HyAR | | | TTPI | | |
|---|---|---|---|---|---|---|---|---|---|---|---|---|---|---|
|  |  |  | $T$ | $\mu$ | $S$ | $T$ | $\mu$ | $S$ | $T$ | $\mu$ | $S$ | $T$ | $\mu$ | $S$ |
| CP | 4 | 2 | 0.5h | 0.13 ±0.01 | 86% ±6% | 1.9h | 0.16 ±0.05 | 84% ±6% | 4h | 0.15 ±0.02 | 92% ±4% | 30s | 1 | 100% |
| HM(8) | 4 | 16 | 1.4h | 0.15 ±0.01 | 8% ±2% | 2.1h | 0.19 ±0.03 | 8% ±3% | 8h | 0.92 ±0.01 | 100% | 850s | 0.93 ±0.01 | 100% |
| HM(12) | 4 | 24 | NA | NA | NA | NA | NA | NA | 10h | 0.92 ±0.05 | 12% ±5% | 946s | 0.92 ±0.01 | 100% |
| HM(16) | 4 | 32 | NA | NA | NA | NA | NA | NA | 10h | NA | 0% | 1743s | 0.92 ±0.02 | 100% |

Table 1: We used the success rate ($S$) for reaching the target position as one of the metrics. We note that the primary objective of both approaches, in the problems considered here, is to reach the goal in the shortest possible time or path. So as a second metric ($\mu$), we calculate the square of the ratio between the length of the trajectory generated by each policy and the length of the shortest path for HM task. For CP task, $\mu$ is the inverse of the number of catch motions till reaching the goal. The table includes the training time ($T$) required to obtain the policy used for evaluation. The number of states is $d$ and the number of actions is $m$.

It is modeled using 6 states and 3 actions. The action includes a discrete variable representing the index of next contact face. Its underactuated and hybrid nature, coupled with multiple discrete contact modes, makes it difficult to design effective control strategies, and it has been a test-bed problem for the control of hybrid systems. Previous approaches, such as mixed integer programming (Hogan & Rodriguez, 2020) and hybrid Differential Dynamic Programming (Doshi et al., 2020), have struggled with the high computational cost required for solving the problem, which requires robust algorithms that can handle the complexity of hybrid systems with both continuous and discrete variables. Note that typically such a non-prehensile manipulation problem is formulated differently as continuous control (Ferrandis et al., 2023), due to a lack of methodologies to handle hybrid actions and is not representative of hybrid control in robotics applications.

Our algorithm achieves robust performance: 100% success rate (reaching the goal) in both simulation and real-world experiments for this task. The experiments demonstrate successful reaching of the target position and orientation, even in the presence of additional weight and external disturbances, as shown in Fig. 3. This indicates the potential of TTPI for solving complex hybrid system control.

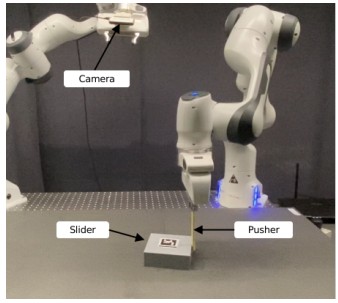 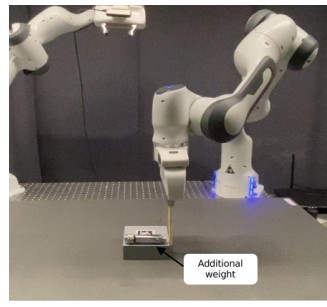 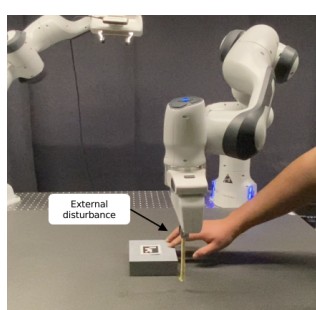

(a) Reaching     (b) Reaching w. additional weight     (c) Reaching w. disturbance

Figure 3: Pusher-slider system where the robot pushes an object by contact switching. Three experiments were performed: The block being pushed towards the target as modeled (a), with additional weight on the block leading to nonuniform friction distribution (b), and with external disturbance (c). The policy obtained from TTPI was robust to handle these scenarios.

## 4 LIMITATION

TTPI approximates the state-value and advantage function over the entire state-action space, resulting in a highly generalizable policy. However, computational complexity and storage issues may arise when these functions are not low-rank in the TT representation. For instance, systems involving highly agile movements like the acrobat (double pendulum swing-up) can lead to high-rank in the TT representation. Nonetheless, decreasing the time step $\Delta t$ has been observed to reduce the rank

of these functions which may enable the approach to handle such systems at the expense of longer training time.

TTPI may be well-suited for commonly encountered systems with discontinuities and hybrid characteristics, such as manipulation and legged robotics. However, a drawback is its reliance on highly parallelized simulators. Hand-coding the system dynamics and reward function, as demonstrated in this paper, may not be practical for more complex dynamics involving contact. While existing simulators like Mujoco and Raisim are not parallelizable and may slow down the process, the availability of recently introduced GPU-based simulators like NVIDIA Isaac Gym presents an opportunity to test the algorithm on more intricate applications.

Concerning scalability, although existing Deep RL techniques struggle to handle hybrid action space, they can cope well with high-dimensional state space (e.g., images as states). On the other hand, TTPI can handle high-dimensional hybrid actions and perform better compared to existing ADP methods, it may not be suitable for very high-dimensional state spaces. However, we could potentially enable our method to handle such high-dimensional problems by formulating our approach as an RL problem instead of ADP or OC. In such cases, instead of TT-Cross, gradient-based methodologies (Novikov et al., 2017; Stoudenmire & Schwab, 2016) could be used to find the TT model of the value and advantage functions. We will investigate this in our future work.

## 5 RELATED WORK

In recent years, research has surged in the domain of optimal control for hybrid systems which involve a mix of discrete and continuous state and action variables. Classical techniques, like Mixed-Integer Programming (MIP) (Marcucci & Tedrake, 2020), unify continuous and discrete variables in a single optimization problem. Abstraction and reachability analysis methods (Alur et al., 2006) help adapt hybrid systems for traditional solvers. However, they often involve high computational complexity and are not suitable for real-time decision-making. This motivates the development of Approximate Dynamic Programming (ADP) techniques, which involve approximating value functions to alleviate computational burdens and handle high-dimensional settings.

The use of low-rank tensor approximation techniques for solving ADP was previously proposed in Horowitz et al. (2014), Gorodetsky et al. (2015), and Boyko et al. (2021). In Gorodetsky et al. (2015) and Boyko et al. (2021), they proposed a TT-based value iteration algorithm, where the TT was used to approximate the value function, and the policy was retrieved using Newton-type optimization technique based on the value function. This limits the application and speed of the algorithm, as the policy retrieval procedure demands the system dynamics and the reward function to be differentiable and the action space to be continuous.

Some of the NN-based ADP for continuous state and action space have been proposed in fitted-Q iteration (Antos et al., 2007) and fitted-value iteration (Lutter et al., 2022). However, these methods have demonstrated their applicability only to systems with low dimensional systems and they have not been successful in handling hybrid action space. The NN-based ADP methods have been overshadowed by the rise of Deep RL as they have demonstrated scalability to problems with high dimensional state and action space. To overcome the issues in Deep RL for handling hybrid actions several improvements were proposed by Hausknecht & Stone (2016); Fu et al. (2019); Fan et al. (2019) and Li et al. (2022).

## 6 CONCLUSION

In this paper, we presented TTPI, an ADP algorithm that can handle hybrid action space. Through simulation experiments, we showed that the algorithm is superior to state-of-the-art algorithms for dealing with hybrid action spaces in terms of training time, generalization, and the quality of the policy. We demonstrated the robustness of the policy of TTPI through real-world experiments. The results demonstrate that our approach could be promising in robotics for solving challenging hybrid control tasks.

# A APPENDICES

## A.1 TT VS NN

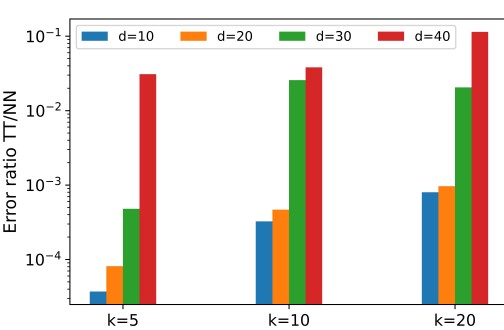

Figure 4: The y-axis represents in log scale the ratio of approximation error in TT representation to that of the NN. The target function for approximation was chosen to be a GMM with various choices of mixture components $k$ and dimensionality $d$. The graph shows the superiority of TT modeling over NN for modeling such functions. For example, the graph shows that for $k = 20$ and $d = 40$, the error with TT is 10 times smaller than with NN. For k=5 and d=10, the error with TT is 10'000 times smaller than with NN.

In this section, we provide a comprehensive analysis of the performance of Tensor Train (TT) compared to Neural Networks (NN) for function approximation. We employed a Gaussian Mixture Model (GMM) as a representative target function, varying the number of mixture components (2 to 20) and dimensionality (2 to 40). The results in Figure 4 demonstrate that TT-Cross could accurately represent the target function in TT format, achieving orders of magnitude better accuracy compared to NN, and requiring significantly less time. Additionally, the nonparametric and unsupervised nature of TT-Cross offers flexibility with minimal intervention, eliminating the need for careful selection, unlike the case of NN which requires careful hyperparameter tuning such as NN architecture, learning rate, and batch size. The superiority of TT-Cross is due to the fact that the algortihm can directly query the function to evaluate trials, while exploiting the low-rank structure for the approximation. This motivates the use of TT for modeling value functions and advantage functions in Approximate Dynamic Programming (ADP).

In our experiment, TT-Cross was able to find the TT representation of the GMM with less than $10^{-6}$ error as specified in TT-Cross in under 20 seconds for each test case, while NN took several minutes and had a significantly higher error (often several orders of magnitude higher). Furthermore, NN required significant effort to tune the hyperparameters, whereas TT-Cross, as it is a non-parametric and unsupervised approach, was much easier to use. This is because TT-Cross finds the approximation by querying data (the function values at various points) intelligently (Savostyanov & Oseledets, 2011) and exploits the structure in the function (i.e., low-rank or separability). It can do so as TT-Cross directly takes the function to be approximated as the input. On the other hand, NN takes a fixed set of samples from the function and does supervised learning to find the function approximation. We acknowledge that the approximation error in NN in our experiments could potentially be reduced by using more training data, and using a more exhaustive search for best hyperparameters. However, this would increase the training time and manual effort.

Although NN is an established tool for supervised learning over datasets, it is inefficient, compared to TT-Cross, when we need to approximate a known low-rank function accurately. Unlike TT-Cross, NN works with data collected from the function for the approximation and does not have a feedback mechanism to query points from the function during the approximation procedure. Thus, choosing NN as a function approximation technique in ADP, where we need to repeatedly approximate value functions from the previous estimations, comes with a drawback. The software code for this comparison is provided in the supplementary material.

## A.2 ADDITIONAL SIMULATION EXPERIMENTS

In addition to the benchmark problems on hybrid actions provided in the main section, we performed further experiments to evaluate the performance of our approach on some benchmark optimal control problems involving continuous states, including Point-mass control with obstacles, Cart-Pole Swing-up, and Box-pivoting. The video provided with the supplementary material shows the performance of the policy obtained by TTPI on these tasks.

A.3   EXPERIMENTS WITH REAL ROBOT: PLANAR PUSHING TASK

In this section, we discuss the performance of our proposed method for a planar pushing task (Xue et al., 2023). It is considered to be challenging in the field of model-based planning and control, due to its underactuated and hybrid nature with several discrete contact modes.

The objective of the task is to push a block with the option of switching the face of the block to be pushed, as well as the contact mode used for pushing. We demonstrate that the proposed algorithms can achieve this task robustly in both simulation and the real world. A video of the experiments is provided in the supplementary material.

The state of the system is denoted as $[\boldsymbol{q}_s^\top \ \boldsymbol{q}_p^\top \ c_c]^\top$, where $\boldsymbol{q}_s = [s_x \ s_y \ s_\theta]^\top$ is the position and orientation of the block, $\boldsymbol{q}_p = [p_x \ p_y]^\top$ is the position of end-effector, and $c_c \in \{0, 1, 2, 3\}$ is the current contact face. The action is expressed as $[\boldsymbol{v}^\top \ c_n]^\top$, where $\boldsymbol{v} = [v_n \ v_t]^\top$ is the velocity of the end-effector, and $c_n \in \{0, 1, 2, 3\}$ is the next contact face. The system, therefore, has $n = 6$ states and $m = 3$ control variables in total, including both continuous and discrete variables.

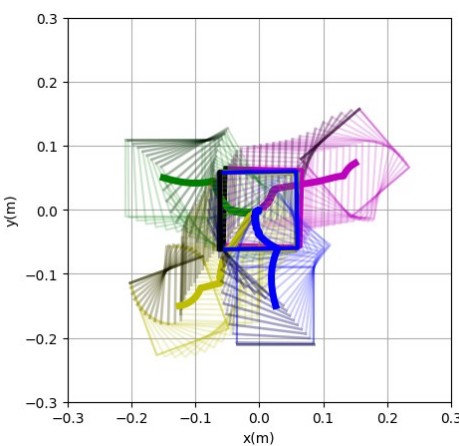

Figure 5: Simulation of the motion of the block under a policy from four different initial states. The colored trajectories represent the motion of the block to the target ($\boldsymbol{q}_s = [0 \ 0 \ 0]^\top$), by means of contact mode and face switching.

We first trained the control policy in simulation based on the predefined motion equation. The continuous variables in state and action spaces are discretized into 100 bins. The domain is set in the range from $[-0.5m, -0.5m, -\pi]$ to $[0.5m, 0.5m, \pi]$, with maximum velocity defined as 0.1 m/s. The accuracy of TT-cross is defined as $10^{-3}$. The rank of the final value function was found to be 4 and the rank of the advantage function was 40. Each iteration of the VI procedure took about 10 seconds on average. To test the generalization capability of the policy, we randomly selected 1000 initialization points in the domain. A success rate of 100% was obtained in under 10 minutes of training. Fig. 5 shows the simulation results. The reward function is defined as

$$R(\boldsymbol{s}, \boldsymbol{a}) = -2\|\boldsymbol{q}_s\| - (1 - \delta(c_c - c_n)), \tag{10}$$

where $\boldsymbol{q}_s$ represent the block pose, $\delta(c_c - c_n)$ will return 1 if $c_c = c_n$ (no face switching), otherwise, 0. Note that the flexibility offered by our method allows us to utilize such reward functions.

We then tested the trained policy on the real robot setup (Fig. 3), using a 7-axis Franka Emika robot and a RealSense D435 camera. The slider ($r_s = 6$ cm) is a 3D-printed prismatic object with PLA, lying on a flat plywood surface, with an Aruco Marker on the top face. A wooden pusher ($r_p = 0.5$cm) is attached to the robot to move the object. The motion of the object is tracked by the camera at 30 HZ, and the policy is updated at 100 HZ, with a low-level Cartesian velocity controller (1000 HZ) actuating the robot.

Three experiments were conducted to assess the robustness of our policy: **a) Reaching task:** The robot pushes the slider from $\boldsymbol{q}_{s_0} = [0.05m \ 0.16m \ 0]^\top$ to the origin (Fig. 3a); **b) Reaching with additional weight:** The robot pushes the block from the same initialization as before, but with an additional weight, 3 times heavier than the block (Fig. 3b); **c) Reaching with external disturbance:** The same initialization like before, but with a significant external disturbance of $\boldsymbol{q}_{\text{dist}} = [0.1m \ 0.03m \ 90°]^\top$ exerted by a human (Fig. 3c).

Table 2: Performance of three real-world experiments

| Experiments | $x_{\text{err}}$/cm | $y_{\text{err}}$/cm | $\theta_{\text{err}}$/rad |
|---|---|---|---|
| Reaching | -0.83 | 1.07 | -0.06 |
| Reaching with additional weight | 2.89 | -1.04 | -0.01 |
| Reaching with external disturbance | -4.78 | -4.10 | -0.04 |

The results of these experiments are shown in Table 2. The results show that in all experiments, the policy successfully reaches the final target in terms of both position and orientation. The error increases with the disturbance, while orientation errors remain less than $4°$ and position errors remain less than 5cm even under significant disturbance. Experiment 3 demonstrates that the policy is able to dynamically select the contact face based on the current state, as evidenced by the change in contact face after a $90°$ rotation. This highlights the ability of our method to handle both continuous and discrete variables in hybrid systems.

### A.4 CROSS APPROXIMATION METHODS

The popular methods to find the TT decomposition of a tensor are TT-SVD (Oseledets, 2011), TT-DMRG (Dolgov & Savostyanov, 2020), and TT-cross (Savostyanov & Oseledets, 2011). TT-SVD and TT-DMRG, like matrix SVD, require the full tensor in memory to find the decomposition, and hence they are infeasible for higher-order tensors. TT-cross approximation is an extension of the matrix cross approximation technique for obtaining the TT decomposition of a tensor. It is appealing for many practical problems as it approximates the given tensor with a controlled accuracy, by evaluating only a small number of its elements and without having to compute and store the entire tensor in the memory. The method needs to compute only certain fibers of the original tensor at a time and hence works in a black-box fashion. In this section, we describe the matrix cross-approximation algorithm to provide an intuition about TT-Cross and we refer the readers to (Sozykin et al., 2022; Oseledets & Tyrtyshnikov, 2010; Savostyanov & Oseledets, 2011) for more detail on how it can be adapted to find the TT decomposition of higher-dimensional tensors using TT-cross.

Suppose we have a rank-$r$ matrix $\boldsymbol{P} \in \mathbb{R}^{n_1 \times n_2}$. Using cross-approximation (a.k.a. CUR decomposition or skeleton decomposition), this matrix can be exactly recovered using $r$ independent rows (given by the index vector $\boldsymbol{i}_1 \subset \{1, \ldots, n_1\}$) and $r$ independent columns (given by the index vector $\boldsymbol{i}_2 \subset \{1, \ldots, n_2\}$) of the matrix $\boldsymbol{P}$ as

$$\hat{\boldsymbol{P}} = \boldsymbol{P}_{:,\boldsymbol{i}_2} \ \boldsymbol{P}_{\boldsymbol{i}_1,\boldsymbol{i}_2}^{-1} \ \boldsymbol{P}_{\boldsymbol{i}_1,:},$$

provided the intersection matrix $\boldsymbol{P}_{\boldsymbol{i}_1,\boldsymbol{i}_2}$ (called submatrix) is non-singular. Thus, the matrix $\boldsymbol{P}$, which has $n_1 n_2$ elements, can be reconstructed using only $(n_1 + n_2 - r)r$ of its elements (see Figure 6).

Now suppose we have a noisy version of the matrix $\boldsymbol{P} = \tilde{\boldsymbol{P}} + \boldsymbol{E}$ with $\|\boldsymbol{E}\| < \epsilon$ and $\tilde{\boldsymbol{P}}$ is of low rank. For a sufficiently small $\epsilon$, $\text{rank}(\tilde{\boldsymbol{P}}) = r$ so that the matrix $\boldsymbol{P}$ can be approximated with a lower rank $r$ (i.e., $\text{rank}(\boldsymbol{P}) \approx r$). Then, the choice of the submatrix $\boldsymbol{P}_{\boldsymbol{i}_1,\boldsymbol{i}_2}$ (or index vectors $\boldsymbol{i}_1, \boldsymbol{i}_2$) for the cross approximation requires several considerations. The maximum volume principle can be used in choosing the submatrix which states that the submatrix with maximum absolute value of the determinant is the optimal choice. If $\boldsymbol{P}_{\boldsymbol{i}_1^*,\boldsymbol{i}_2^*}$ is chosen to have the maximum volume, then by skeleton decomposition we have an approximation of the matrix $\boldsymbol{P}$ given by $\hat{\boldsymbol{P}} = \boldsymbol{P}_{:,\boldsymbol{i}_2^*} \boldsymbol{P}_{\boldsymbol{i}_1^*,\boldsymbol{i}_2^*}^{-1} \boldsymbol{P}_{\boldsymbol{i}_1^*,:}$. This results in a quasi-optimal approximation

$$\|\boldsymbol{P} - \hat{\boldsymbol{P}}\|_2 < (r+1)^2 \ \sigma_{r+1}(\boldsymbol{P}),$$

where $\sigma_{r+1}(\boldsymbol{P})$ is the $(r+1)$-th singular value of $\boldsymbol{P}$ (i.e., the approximation error in the best rank $r$ approximation in the spectral norm). Thus, we have an upper bound on the error incurred in the approximation which is slightly higher than the best rank $r$ approximation (Eckart–Young–Mirsky theorem).

Finding the maximum volume submatrix is, however, an NP-hard problem. However, many heuristic algorithms that work well exist in practice by using a submatrix with a sufficiently large volume, trading off the approximation accuracy for the computation speed. One of the widely used methods is the MAXVOL algorithm (Goreinov et al., 2010) which can provide, given a tall matrix $\boldsymbol{P} \in \mathbb{R}^{r \times n_2}$ (or $\mathbb{R}^{n_1 \times r}$), the maximum volume submatrix $\boldsymbol{P}_{\boldsymbol{i}_1^*,\boldsymbol{i}_2^*} \in \mathbb{R}^{r \times r}$. The cross approximation algorithm uses the MAXVOL algorithm in an iterative fashion to find the skeleton decomposition as follows:

1. *Input*: $\boldsymbol{P} \in \mathbb{R}^{n_1 \times n_2}$, the approximation rank $r$ for the skeleton decomposition.

2. Find the columns index set $\boldsymbol{i}_2^*$ and the row index set $\boldsymbol{i}_1^*$ corresponding to the maximum volume submatrix.

    2.1 Randomly choose $r$ columns $\boldsymbol{i}_2$ of the matrix $\boldsymbol{P}$ and repeat the following until convergence:
   - Use MAXVOL to find $r$ row indices $\boldsymbol{i}_1$ so that $\boldsymbol{P}_{\boldsymbol{i}_1, \boldsymbol{i}_2}$ is the submatrix with maximum volume in $\boldsymbol{P}_{:, \boldsymbol{i}_2}$.
   - Use MAXVOL to find $r$ column indices $\boldsymbol{i}_2$ so that $\boldsymbol{P}_{\boldsymbol{i}_1, \boldsymbol{i}_2}$ is the submatrix with maximum volume in $\boldsymbol{P}_{\boldsymbol{i}_1, :}$.

3. *Output*: Using the column index set $\boldsymbol{i}_2^*$ and the row-index set $\boldsymbol{i}_1^*$ corresponding to the maximum volume submatrix, we have the skeleton decomposition $\hat{\boldsymbol{P}} \approx \boldsymbol{P}_{:, \boldsymbol{i}_2^*} \boldsymbol{P}_{\boldsymbol{i}_1^*, \boldsymbol{i}_2^*}^{-1} \boldsymbol{P}_{\boldsymbol{i}_1^*, :}$.

In the above algorithm, during the iterations, the matrices $\boldsymbol{P}_{:, \boldsymbol{i}_2}$ (or $\boldsymbol{P}_{\boldsymbol{i}_1, :}$) might be singular. Thus, a more practical implementation uses the pseudoinverses of these matrices. For details on practical implementation, we refer to Kishore Kumar & Schneider (2017). Note that, in the above algorithm, the input is only a function to evaluate the elements of the matrix $\boldsymbol{P}$ (i.e., we do not need the whole matrix $\boldsymbol{P}$ in computer memory).

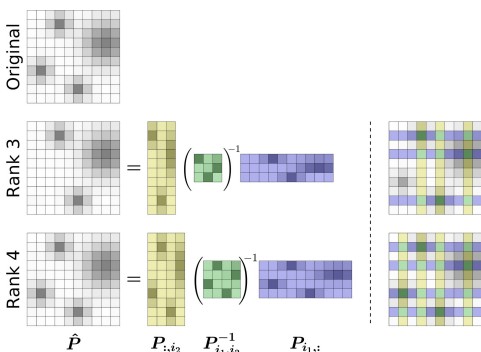

Figure 6: For a given matrix $\boldsymbol{P}$ (top-left), suppose we know $r$ independent columns indexed by $\boldsymbol{i}_2 = (i_{2,1}, \ldots, i_{2,r})$, i.e., $\boldsymbol{P}_{:, \boldsymbol{i}_2} \in \mathbb{R}^{n_1 \times r}$ and $r$ independent rows indexed by $\boldsymbol{i}_1 = (i_{1,1}, \ldots, i_{1,r})$, i.e., $\boldsymbol{P}_{\boldsymbol{i}_1, :} \in \mathbb{R}^{r \times n_2}$, with their intersection $\boldsymbol{P}_{\boldsymbol{i}_1, \boldsymbol{i}_2} \in \mathbb{R}^{r \times r}$ being nonsingular. Then, by skeleton decomposition we have $\hat{\boldsymbol{P}} = \boldsymbol{P}_{:, \boldsymbol{i}_2} \boldsymbol{P}_{\boldsymbol{i}_1, \boldsymbol{i}_2}^{-1} \boldsymbol{P}_{\boldsymbol{i}_1, :}$.

If $\text{rank}(\boldsymbol{P}) = r$, then $\hat{\boldsymbol{P}} = \boldsymbol{P}$ (bottom row). For $r < \text{rank}(\boldsymbol{P})$ we obtain a quasi-optimal approximation, $\hat{\boldsymbol{P}} \approx \boldsymbol{P}$ (middle row). The right figures show the rows and columns selected from the original matrix by the cross-approximation algorithm to find the skeleton decomposition.

### A.5 REFINING TENSOR TRAIN MODEL

Suppose we have a TT model $\mathcal{P}$ defined on a domain $\Omega_{\boldsymbol{x}} = \Omega_{x_1} \times \cdots \times \Omega_{x_d}$ with discretization set $\mathcal{X} = \left\{ \boldsymbol{x} = (x_1^{i_1}, \ldots, x_d^{i_d}) : x_k^{i_k} \in \Omega_{x_k}, \ i_k \in \{1, \ldots, n_k\} \right\}$. We can obtain a refined TT model $\hat{\mathcal{P}}$ defined on a finer discretization $\hat{\mathcal{X}} = \left\{ \boldsymbol{x} = (x_1^{i_1}, \ldots, x_d^{i_d}) : x_k^{i_k} \in \Omega_{x_k}, i_k \in \{1, \ldots, \hat{n}_k\} \right\}$ of the domain with $\hat{n}_k > n_k$ using interpolation of the TT cores in a fast manner. The cores of the corresponding TT model $\hat{\mathcal{P}}$ defined over the refined discretization can be determined using $\hat{\mathcal{P}}_{:, x_k, :}^k = \boldsymbol{P}^k(x_k), \ \forall k \in \{1, \ldots, d\}, \ (x_1, \ldots, x_d) \in \hat{\mathcal{X}}$ using spline-based interpolation scheme as described in Section 2.6.

This proves beneficial in specific applications where a coarse discretization is employed during the learning phase to acquire the TT representation of a target function, which can be computationally intensive. During the inference phase, however, in some applications, the TT model might be required to be defined over a finer discretization for more accurate results. For instance, in our work, while modeling the advantage function using TT-cross we use a coarser discretization. However, the advantage function in TT format used for computing the policy using TTGO uses a finer discretization. This ensures the attainment of a more accurate optimum over action space.

### A.6 TENSOR TRAIN DISTRIBUTION

Suppose we have a tensor $\mathcal{P}$ in TT format corresponding to a function $P$ with the discretization set $\mathcal{X}$ of the domain $\Omega_{\boldsymbol{x}}$. We can then construct the corresponding probability distribution that we call TT distribution,

$$\text{Pr}(\boldsymbol{x}) = \frac{\mathcal{P}_{\boldsymbol{x}}^2}{Z}, \quad \boldsymbol{x} \in \mathcal{X}, \tag{11}$$

where $Z$ is the corresponding normalization constant.

### A.7 Conditioning Tensor Train Distribution

Suppose we want to fix a subset of variables in $x$ and find the corresponding conditional distribution of the remaining variables. Without loss of generality, let $x$ be segmented as $x = (x_1, \ x_2) \in \Omega_x = \Omega_{x_1} \times \Omega_{x_2}$ with $x_1 \in \Omega_{x_1} \subset \mathbb{R}^{d_1}, \ x_2 \in \Omega_{x_2} \subset \mathbb{R}^{d_2}$. i.e., $x_1$ corresponds to the first $d_1$ variables in $x$. We are interested in finding the conditional distribution $\Pr(x_2|x_1)$ of the TT distribution given in (11).

Suppose $x_1$ takes a particular value $x_t = (x_1, \ldots, x_{d_1})$. We can obtain $\Pr(x_2|x_1 = x_t)$ by defining a conditional TT model $\mathcal{P}^{x_1=x_t}$ using TT model $\mathcal{P}$ as

$$\mathcal{P}^{x_t}_{x_2} = \mathcal{P}_{(x_t, \ x_2)}, \quad \forall x_2 \in \Omega_{x_2}.$$

In other words, the TT cores of $\mathcal{P}^{x_1=x_t}$ are then given by

$$(\mathcal{P}^{x_t})^1_{:, \ x, \ :} = \Big( \prod_{i=1}^{d_1} \mathcal{P}^i_{:, \ x_i, \ :} \Big) \mathcal{P}^{d_1+1}_{:, \ x, \ :}, \quad \forall x \in \Omega_{x_{d+1}}, \tag{12}$$

$$(\mathcal{P}^{x_t})^k = \mathcal{P}^{k+d_1}, \ k \in \{2, \ldots, d_2\}.$$

Given the above-defined conditional TT model, we can obtain the conditional distribution as

$$\Pr(x_2|x_1 = x_t) = \frac{(\mathcal{P}^{x_t}_{x_2})^2}{Z_2}, \quad \forall x_2 \in \Omega_{x_2}. \tag{13}$$

In this work, $x_1$ is the state variable, $x_2$ is the action variable, and $P$ corresponds to a transformed advantage function (see Section A.11).

### A.8 Sampling from Tensor Train distribution

Consider the discrete probability distribution given by (11). For the simplicity of the presentation, we assume $Z = 1$ as we will not require the normalization constant to be known for sampling from the above distribution. Any probability distribution can be expressed as a product of conditional distributions

$$\Pr(x_1, \ldots, x_d) = \Pr_1(x_1) \Pr_2(x_2|x_1) \cdots \Pr_d(x_d|x_1, \ldots, x_{d-1}),$$

where

$$\Pr_k(x_k|x_1, \ldots, x_{k-1}) = \frac{\sigma_k(x_1, \ldots, x_k)}{\sigma_{k-1}(x_1, \ldots, x_{k-1})},$$

is the conditional distribution defined using the marginals

$$\sigma_k(x_1, \ldots, x_k) = \sum_{x_{k+1}} \cdots \sum_{x_d} \Pr(x_1, \ldots, x_d).$$

Let $\sigma_0 = 1$. Now, using the above definitions, we can generate samples $x \sim \Pr$ by sampling from each of the conditional distributions in turn. Each conditional distribution is a function of only one variable, and in the discrete case it is a multinomial distribution with

$$x_k \sim \Pr_k(x_k|x_1, \ldots, x_{k-1}), \ \forall k \in \{1, \ldots, d\}.$$

However, this process becomes computationally intensive during sampling $x_k$, as it necessitates the conditional distribution $\Pr_k$, which, in turn, requires the evaluation of the summation over several variables to find the marginal $\sigma_k$. Consequently, this approach incurs a computational cost that grows exponentially with the number of dimensions. Here, the TT format offers an elegant solution by capitalizing on the separability of the function.

If $\Pr$ is a TT distribution (see (11)) corresponding to a TT model $\mathcal{P}$ with the discretization set $\mathcal{X}$ and the cores $(\mathcal{P}^1, \ldots, \mathcal{P}^d)$, we have

$$\sigma_k(x_1, \ldots, x_k) = \sum_{x_{k+1}} \cdots \sum_{x_d} \mathcal{P}^2_x, \quad k \in \{1, \ldots, d\}$$

$$= \Big( \mathcal{P}^1_{:, \ x_1, \ :} \cdots \mathcal{P}^k_{:, \ x_k, \ :} \Big) \beta^{k+1} \Big( \mathcal{P}^1_{:, \ x_1, \ :} \cdots \mathcal{P}^k_{:, \ x_k, \ :} \Big)^\top, \tag{14}$$

---

**Algorithm 3** Sampling from TT distribution.

1: **Input:** TT Blocks $\mathcal{P} = (\mathcal{P}^1, \ldots, \mathcal{P}^d)$ corresponding to the distribution Pr, sample priority $\alpha \in (0, 1)$
2: **Output:** $N$ samples $\{(x_1^l, \ldots, x_d^l)\}_{l=1}^N$ from the distribution Pr (see (11))
3: $\beta^{d+1} \leftarrow 1$
4: **for** $k \leftarrow d$ to 2 **do**
5: $\quad \beta^k = \sum_{x_k} \mathcal{P}^k_{:, x_k, :} \beta^{k+1} (\mathcal{P}^k_{:, x_k, :})^\top$
6: **end for**
7: $\Phi_1 \leftarrow \mathbf{1} \in \mathbb{R}^{N \times 1}$
8: **for** $k \leftarrow 1$ to $d$ **do**
9: $\quad \pi^k(x_k) = \mathcal{P}^k_{:, x_k, :} \beta^{k+1} (\mathcal{P}^k_{:, x_k, :})^\top, \quad \forall x_k$
10: $\quad$ **for** $l = 1, \ldots, N$ **do**
11: $\quad\quad p_k(x_k) = |\Phi_k(l, :) \pi^k(x_k) \Phi_k(l, :)^\top|, \quad \forall x_k$
12: $\quad\quad$ Sample $x_k^l$ from the multinomial distribution $p_k$
13: $\quad\quad \Phi_{k+1}(l, :) = \Phi_k(l, :) \mathcal{P}^k_{:, x_k^l, :}$
14: $\quad$ **end for**
15: **end for**

---

where we can compute $\beta^k$ efficiently in a recursive manner as

$$
\begin{aligned}
\beta^k &= \sum_{x_k} \cdots \sum_{x_d} \left( \mathcal{P}^k_{:, x_k, :} \cdots \mathcal{P}^d_{:, x_d, :} \right) \left( \mathcal{P}^k_{:, x_k, :} \cdots \mathcal{P}^d_{:, x_d, :} \right)^\top \\
&= \sum_{x_k} \mathcal{P}^k_{:, x_k, :} \beta^{k+1} (\mathcal{P}^k_{:, x_k, :})^\top, \quad k \in \{d, \ldots, 2, 1\},
\end{aligned}
\tag{15}
$$

where $\beta^{d+1} = 1$. Alternatively, there exists a procedure called Tensor Train orthogonalization (Lee & Cichocki, 2018) which re-parameterizes the core tensors of $\mathcal{P}$ in TT format so that $\beta^k$ is an identity matrix. This will eliminate the need for the computation given in (14). Thus, the TT format reduces the complicated multidimensional summation to evaluate $\sigma_k$ into several one-dimensional summations. As the same summation terms appear over several conditionals $\text{Pr}_k$, we can obtain an efficient algorithm to sample from the TT distribution Pr which is described in Algorithm 3. For more details, we refer to Dolgov et al. (2020).

### A.9 TTGO: TENSOR TRAIN FOR GLOBAL OPTIMIZATION

In this section, we propose efficient methodologies to find the maxima of a TT-distribution given by (11). In the next sections we show how we can generalize it to find the optima of arbitrary functions in TT format.

The methodology was proposed originally by Shetty et al. (2023) as a stochastic procedure. It was originally introduced to obtain multiple solutions and global optimality of TT distribution. Following this work, a deterministic version of this approach was proposed by Chertkov et al. (2022). In the section, we further improve this deterministic version for applicability in robotics for policy learning.

The idea behind TTGO is to leverage the Algorithm 3 to find the points corresponding to the high-density regions of the TT distribution. Recall that the sampling procedure in Algorithm 3 consists of repeated sampling of each dimension separately from a multinomial distribution. We modify this by prioritizing the high-density values. To do this, instead of sampling $N$ elements independently, we select the top $N$ elements from the multinomial distribution $p_k$ at iteration $k$ and we keep track of the history of the selected indices from the previous modes $(k - 1)$. i.e., we only retain top-N elements of $\text{Pr}_k$. This deterministic version of TTGO was proposed by Chertkov et al. (2022). We further improve this methodology to include smarter choices for top-N which improves the quality and diversity of the solution obtained. The idea is to give higher priority to the local maxima (peaks) of the multinomials involved in each iteration (i.e., $\text{Pr}_k(x_k | x_1, \ldots, x_{k-1}), \forall k \in \{1, \ldots, d\}$ as described in Section A.8). In addition, we introduce an iterative procedure to improve the solution and scalability of the approach. This is sketched in Algorithm 4 and we provide a fully parallel implementation in the software accompanying the paper.

---

**Algorithm 4** Deterministic TTGO

1: **Input:** TT Cores $\boldsymbol{\mathcal{P}} = (\boldsymbol{\mathcal{P}}^1, \ldots, \boldsymbol{\mathcal{P}}^d)$, Domain $\Omega_x = \{(x_1^{i_1}, \ldots, x_d^{i_d}) : i_k \in \{1, \ldots, n_k\}\}$
2: **Hyperparameters:** $N$, $n_{\textbf{sweeps}} \in \{1, \ldots, d\}$, $\epsilon$ # default: $K = n_1$, $n_{\text{sweeps}} = 1$, $\epsilon = 0.001$
3: **Output:** Maxima $\boldsymbol{x}^* = (x_1^*, \ldots, x_d^*)$ of the TT distribution given by $\boldsymbol{\mathcal{P}}$ (see (11))
4: $\boldsymbol{\beta}^{d+1} \leftarrow 1$
5: **for** $k \leftarrow d$ to 2 **do**
6: $\quad \boldsymbol{\beta}^k = \sum_{x_k} \boldsymbol{\mathcal{P}}^k_{:, \, x_k, \, :} \, \boldsymbol{\beta}^{k+1} \, \boldsymbol{\mathcal{P}}^k_{:, \, x_k, \, :}{}^\top$
7: **end for**
8: **Definition:**

$$\pi^m(x_1^{i_1}, \ldots, x_m^{i_m}) = (\boldsymbol{\mathcal{P}}^1_{:, \, x_1^{i_1}, \, :} \cdots \boldsymbol{\mathcal{P}}^m_{:, \, x_m^{i_m}, \,})\boldsymbol{\beta}^{m+1}(\boldsymbol{\mathcal{P}}^1_{:, \, x_1^{i_1}, \, :} \cdots \boldsymbol{\mathcal{P}}^m_{:, \, x_m^{i_m}, \,})^\top$$

$$q^m(x_1^{i_1}, \ldots, x_m^{i_m}) = \begin{cases} 1 & \text{if} \quad \begin{matrix} (\pi^m(x_1^{i_1}, \ldots, x_{m-1}^{i_{m-1}}, \, x_m^{i_m}) > \pi^m(x_1^{i_1}, \ldots, x_{m-1}^{i_{m-1}}, \, x_m^{i_m+a}), \\ \forall a \in \{1, -1\}) \text{ OR } (x_m \text{ is discrete}) \end{matrix} \\ \epsilon & \text{else} \text{ \# i.e., lower weight if } \pi^m \text{ is not a concave peak w.r.t. } x_m^{i_m} \end{cases}$$

$$\hat{\pi}^m(x_1^{i_1}, \ldots, x_m^{i_m}) = q^m(x_1^{i_1}, \ldots, x_m^{i_m}) \, \pi^m(x_1^{i_1}, \ldots, x_m^{i_m})$$

9: **Initialize:** $\mathcal{D}_1^1 = \{(x_1^{j_1^k}) : k \in \{1, \ldots, \min(N, n_1)\}, \, j_i^k \in \{1, \ldots, n_1\}, \, \pi^1(x_1^{j_1^k}) \geq \pi^1(x_1^{j_1^{k-1}})\}$
10: Set $p_{max} = 0$
11: **for** $s \leftarrow 1$ to $n_{\text{sweeps}}$ **do**
12: $\quad$ **for** $m \leftarrow \max(2, s)$ to $d$ **do**
13: $\qquad \mathcal{D}_m^s = \{(x_1^{j_1^k}, \ldots, x_m^{j_m^k}) :$
$\qquad\qquad\qquad k \in \{1, \ldots, \min(N, \, \text{size}(\mathcal{D}_{m-1}^s) \, n_m)\},$
$\qquad\qquad\qquad j_i^k \in \{1, \ldots, n_1\},$
$\qquad\qquad\qquad \hat{\pi}^m(x_1^{j_1^k}, \ldots, x_m^{j_m^k}) \geq \hat{\pi}^m(x_1^{j_1^{k-1}}, \ldots, x_m^{j_m^{k-1}}),$
$\qquad\qquad\qquad (x_1^{j_1^k}, \ldots, x_{m-1}^{j_{m-1}^k}) \in \mathcal{D}_{m-1}^s)\}$
14: $\quad$ **end for**
15: $\quad \boldsymbol{x} = (x_1, \ldots, x_d) \leftarrow (x_1^{j_1^1}, \ldots, x_d^{j_d^1}) \in \mathcal{D}_d^s$
16: $\quad p = |\pi^d(\boldsymbol{x})|$
17: $\quad$ **if** $p \geq p_{max}$ **then**
18: $\qquad p_{max} \leftarrow p$
19: $\qquad \boldsymbol{x}^* \leftarrow \boldsymbol{x}$
20: $\quad$ **end if**
21: $\quad \mathcal{D}_s^s = \{(x_1^*, \ldots, x_s^*)\}$
22: **end for**
23: Note: The associated software provides a highly parallel implementation of the above algorithm in PyTorch where $\mathcal{D}$ are tensors.

---

### A.10 FINDING OPTIMA OF ARBITRARY TENSOR TRAIN MODEL

Given a TT model $\boldsymbol{\mathcal{P}}$, Algorithm 4 provides maxima $\arg\max_{\boldsymbol{x}} |\boldsymbol{\mathcal{P}}(\boldsymbol{x})|$ (i.e. approximation to maximum of the corresponding TT distribution given by (11)). To find the $\arg\max_{\boldsymbol{x}} \boldsymbol{\mathcal{P}}(\boldsymbol{x})$ using TTGO, we first need to pre-process the TT model. We first find the maxima w.r.t the absolute value $\boldsymbol{x}_a = \arg\max_{\boldsymbol{x}} |\boldsymbol{\mathcal{P}}(\boldsymbol{x})|$ which can be done using TTGO with $\boldsymbol{\mathcal{P}}$. Next, we find a shifted TT model $\hat{\boldsymbol{\mathcal{P}}} = \boldsymbol{\mathcal{P}} - \boldsymbol{\mathcal{P}}(\boldsymbol{x}_a)$ (using algebraic operations over TT model). Now we again use TTGO to find $\boldsymbol{x}_b = \arg\max_{\boldsymbol{x}} |\hat{\boldsymbol{\mathcal{P}}}(\boldsymbol{x})|$. $\boldsymbol{x}_a$ and $\boldsymbol{x}_b$ are the two extrema (a maxima and a minima) of $\boldsymbol{\mathcal{P}}$. Thus, $\boldsymbol{x}_{\min} = \arg\min_{\boldsymbol{x} \in \{\boldsymbol{x}_a, \, \boldsymbol{x}_b\}} \boldsymbol{\mathcal{P}}(\boldsymbol{x})$ and $\boldsymbol{x}_{\max} = \arg\max_{\boldsymbol{x} \in \{\boldsymbol{x}_a, \, \boldsymbol{x}_b\}} \boldsymbol{\mathcal{P}}(\boldsymbol{x})$.

### A.11 NORMALIZING TENSOR TRAIN MODEL

Let $\boldsymbol{x}_{\min}$ and $\boldsymbol{x}_{\max}$ be the minima and maxima of an arbitrary TT model $\boldsymbol{\mathcal{P}}$ found using the methodology described in Section A.10. Let $p_{\min} = \boldsymbol{\mathcal{P}}(\boldsymbol{x}_{\min})$ and $p_{\max} = \boldsymbol{\mathcal{P}}(\boldsymbol{x}_{\max})$. Then, we can shift and scale the TT model $\boldsymbol{\mathcal{P}}$ to obtain a new TT model: $\hat{\boldsymbol{\mathcal{P}}} = \hat{p}_{\min} + (\boldsymbol{\mathcal{P}} - p_{\min})\left(\frac{\hat{p}_{\max} - \hat{p}_{\min}}{p_{\max} - p_{\min}}\right)$.

Then, $\hat{\boldsymbol{\mathcal{P}}}(\boldsymbol{x}) \in (\hat{p}_{\min}, \, \hat{p}_{\max})$, $\forall \boldsymbol{x} \in \Omega_{\boldsymbol{x}}$. Note that $\boldsymbol{x}_{\min}$ and $\boldsymbol{x}_{\max}$ are also the extrema of $\hat{\boldsymbol{\mathcal{P}}}$. By specifying $\hat{p}_{\min} > 0$ and $\hat{p}_{\max} > \hat{p}_{\min}$, we can ensure that $\hat{\boldsymbol{\mathcal{P}}}$ is non-negative.

This is a useful pre-processing step in practice to apply TTGO. To find the maxima of an arbitrary tensor $\mathcal{P}$, we can work with the corresponding non-negative TT model $\hat{\mathcal{P}}$. For instance, as we describe later in this chapter, a typical use case of TTGO in robotics is to find $\boldsymbol{x}_d^* = \underset{\boldsymbol{x}_d}{\arg\max}\, \mathcal{P}(\boldsymbol{x}_t,\ \boldsymbol{x}_d)$

where $\mathcal{P}(\boldsymbol{x}_t,\ \boldsymbol{x}_d)$ could be negative. So, we first find the normalized TT model $\hat{\mathcal{P}}$ and then $\boldsymbol{x}_d^* = \underset{\boldsymbol{x}_d}{\arg\max}\, \hat{\mathcal{P}}(\boldsymbol{x}_t,\ \boldsymbol{x}_d)$ can be found using TTGO for various $\boldsymbol{x}_t$ on the conditioned TT model $\hat{\mathcal{P}}_{\boldsymbol{x}_d}^{\boldsymbol{x}_t}$ (as defined in Section A.7). So, unless otherwise specified, we assume the TT model $\mathcal{P}$ is normalized to be non-negative while using TTGO.

### A.12 POLICY COMPUTATION USING TTGO
Recall that TTPI represents the advantage function in TT format obtained using TT-Cross. The policy at each iteration given by $\pi(\boldsymbol{s}) = \underset{\boldsymbol{a}}{\arg\max}\, A(\boldsymbol{s}, \boldsymbol{a})$ is computed using TTGO. Suppose $\mathcal{A}$ denotes the TT representation of the advantage function. In practice, we first normalize the TT model $\mathcal{A}$ to get a normalized TT model $\hat{\mathcal{A}}$ as described in Section A.11. Then given any state $\boldsymbol{s}$, we can compute the policy $\pi(\boldsymbol{s}) = \underset{\boldsymbol{a}}{\arg\max}\, \hat{\mathcal{A}}(\boldsymbol{s},\ \boldsymbol{a})$ using TTGO. More specifically, it is done by applying TTGO on the conditioned TT model $\hat{\mathcal{A}}_{\boldsymbol{a}}^{\boldsymbol{s}}$ as defined in Section 12.

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
