# OpenReview forum: "Generalized Policy Iteration using Tensor Approximation for Hybrid Control"
_ICLR.cc/2024/Conference — ICLR 2024 spotlight_

### Official Review · Reviewer_F3ze · 2023-10-26

**Soundness:** 4 excellent
**Presentation:** 3 good
**Contribution:** 4 excellent
**Rating:** 8
**Confidence:** 4

**Summary:**

The paper proposes a method for solving optimal control problems (assuming a correct dynamics model is available) with mixed continuous and discrete state variables, using an efficient representation of the value function of the problem based on a set of low-dimensional tensors, in a tensor-train (TT) format. An empirical evaluation demonstrates the effectiveness of the proposed method on several control problems of moderate complexity.

**Strengths:**

One of the main strength of the paper lies in the efficient way of computing the optimal action from the tensor-based representation of the value function. It is based on an explicit representation of the advantage function, again in tensor-train format, and the use of the TT-Cross algorithm for an efficient TT approximation of the advantage function after it has been computed by means of a Bellman back-up.

Another strength of the paper is the rather impressive verification of the algorithm on control problem with six continuous state variables and a single discrete variable, on a real robot.

**Weaknesses:**

Although the example on the real robot is very impressive, the examples in simulation are less so. Four-dimensional state space is not that high, barely beyond what can be represented with a look-up table on modern computers. (10^8 cells will take around 400MB of FP numbers.) The authors clearly state that their algorithm is not meant to approximate value functions on very high-dimensional state spaces, such as images, but most robotics applications on 6 degree of freedom robots have 12-dimensional state space, so this is perhaps the dimensionality of highest interest.

Some claims are not entirely justified. For example, the authors say "OC is synonymous with ADP". There is some overlap, but the two fields are hardly the same. Many OC algorithms, including the celebrated LQR and LQG algorithms, are based on basic DP, nothing approximate about it.

Furthermore, the authors say that MuJoCo is not parallelizable. I cannot agree with this, MuJoCo has always been easy to parallelize on multiple CPU cores, and the latest release of MuJoCo, 3.0.0, can run on GPUs and TPUs. True, it came out after ICLR papers were submitted, but please reconsider this claim.

The authors also mention "the need for differentiability of the system dynamics and reward function which is a common assumption in the existing ADP algorithms". This is probably not entirely correct, as many ADP algorithms simply sample the system dynamics.

Furthermore, the authors often use the phrase "policy retrieval", implying that somehow the policy has been lost. Suggest replacing with "policy computation" or "policy evaluation".

Some minor typos:
Appendix A.3: citation missing in the first sentence
Same place: "n=6 states" -> "n=6 state" (completely changes the meaning)
"HZ" -> "Hz"

**Questions:**

How well will the algorithm perform on a somewhat higher dimensional problem, for example a 6 or 12 dimensions?

---

> ### Author Response · Authors · 2023-11-16
> **Response to Reviewer F3ze**
>
> We greatly appreciate your valuable feedback and suggestions and include them in the paper. Here, we offer responses to specific questions and concerns that you have thoughtfully presented.
>
> > *Although the example on the real robot is very impressive, the examples in simulation are less so. Four-dimensional state space is not that high, barely beyond what can be represented with a look-up table on modern computers. (10^8 cells will take around 400MB of FP numbers.) The authors clearly state that their algorithm is not meant to approximate value functions on very high-dimensional state spaces, such as images, but most robotics applications on 6 degree of freedom robots have 12-dimensional state space, so this is perhaps the dimensionality of highest interest.*
> >
>
> > *How well will the algorithm perform on a somewhat higher dimensional problem, for example, a 6 or 12 dimensions?*
> >
>
> Indeed the dimension of the state-space in the benchmark simulation examples for hybrid control is rather low (4D). However, the benchmark simulation problems are representative of high-dimensional action space problems and it is quite large (up to 32D).  In general, in our approach what matters the most is the dimension of the state-action space. We believe the crucial aspect is that TTPI can handle high-dimensional action space which is difficult for existing ADP algorithms.
>
>  In the software library provided, we have included the regulation problem of the point-mass system with obstacles. We have tested it for various dimensionality as it easily forms a testbed for high-dimensional state space. Here the state space is $2m$  where $m$ is the dimension of the action space (acceleration of the point-mass which is the same as the dimension of the environment it lives in). We have tested it for $m = 2, 3, 4, 5$  (so state space of size $4D$  to $10D$) and observed its performance matches the classical algorithms. So, we believe TTPI should be able to handle a relatively larger state space dimension of interest in robotics problems. We will include more examples of robotics problems in the future release of the software.

---

> ### Comment · Reviewer_F3ze · 2023-11-16
> **Response to authors' clarification about state space dimensionality**
>
> I am satisfied with the authors' clarification about state space dimensionality - the joint dimensionality of the state-action space is indeed high enough to make it impossible to use a tabular representation, and the benefit of using the tensor-train representation are clear.

---

> > ### Author Response · Authors · 2023-11-23
> > **Thanks for the Response**
> >
> > We thank the reviewer for the remarks and suggestions!

---

### Official Review · Reviewer_3rMV · 2023-10-31

**Soundness:** 4 excellent
**Presentation:** 4 excellent
**Contribution:** 4 excellent
**Rating:** 8
**Confidence:** 5

**Summary:**

The authors use the Tensor Train (TT) model to approximate the state-value function $V$ and advantage function $A$ during policy iteration while solving optimal control problem. At each step of the policy iteration this functions is being rebuilt using well-known _TT-cross_ algorithm which adaptively queries a certain number of points to the black-box model function to be approximated. In addition, policy $\pi$ is built on these iterations using the known TTGO algorithm, which searches for the maximum of the TT-tensor based on sampling from it. The performance of the algorithm was tested on several model and real robot examples, showing its superiority on them. The distinctive feature of the algorithm is the use of hybrid actions.

**Strengths:**

- The paper is very well structured, the authors have described not only the method and practical application, but also the limitations.

- There is code, which allows for reproducible experimentation. In the supplementary materials there is a video with demonstrations of both synthetic experiments and real experiments with a mechanical arm.

- Numerical and in-situ experiments show the superiority of this method.


- Potentially, this approach is applicable to rather multi-dimensional problems (with large dimensionality of stats or actions) since the TT construction overcomes the curse of dimensionality, and the authors use TT-format compression for all possible functions.

- The approach presented in the paper allows for serious expansion in both quality and time. The authors have identified some of these potential opportunities in the paper.

**Weaknesses:**

I found no significant weaknesses in this article.
As a small remark, the use of a uniform grid in section 2.6 could be pointed out, while it might be more accurate to use, for example, a Chebyshev grid.
Also there are no theoretical estimates (ranks, for example) and no discussion when we expect small TT-ranks. I.e., can we say in advance, without iterating, that the method works. However, this is a general problem of methods using a low-parameter tensor representation.

The paper cited as arXiv (Sozykin et al., 2022) is now officially published: https://proceedings.neurips.cc/paper_files/paper/2022/hash/a730abbcd6cf4a371ca9545db5922442-Abstract-Conference.html

maybe typo: p4 "decomposition equation 5"

**Questions:**

- did you use ideas from _opitma_tt_ (paper Chertkov et al. (2022)) which utilize _top-k_ scheme and thus more accurate?
- what are the typical TT-ranks for double pendulum swing-up in the case of small $\Delta t$ when the model does work?

---

> ### Author Response · Authors · 2023-11-16
> **Response to Reviewer 3rMV**
>
> We greatly appreciate your valuable feedback and suggestions. We also appreciate your suggestions about other discretization procedures that we could employ. Here, we offer responses to specific questions that you have thoughtfully presented.
>
> > *did you use ideas from opitma_tt (paper Chertkov et al. (2022)) which utilize top-k scheme and thus more accurate?*
> >
>
> As mentioned in the paper, we have included the ideas from optima_tt (which is the deterministic version of TTGO). The main advantage is that the policy iteration is stable due to the deterministic nature of the policy.
>
> > *what are the typical TT-ranks for double pendulum swing-up in the case of small when the model does work?*
> >
>
> For the cartpole swingup and double pendulum experiments, we had to use a time step lower than 1 ms. This resulted in a TT rank of the advantage function between 50 to 100 over the policy iteration stages while the value function had a rank was about 50.

---

> > ### Comment · Area_Chair_f8zX · 2023-11-22
> > **Please respond to the author reply**
> >
> > Dear reviewer, please do respond to the author reply and let them know if this has answered your questions/concerns.

---

> > ### Comment · Reviewer_3rMV · 2023-11-22
> >
> > I thank the authors for the answers to my questions. I keep my positive score (8: accept, good paper) and increase confidence in it.

---

> > > ### Author Response · Authors · 2023-11-23
> > > **Thanks for reviewer's response**
> > >
> > > We thank the reviewer for the positive feedback!

---

### Official Review · Reviewer_ntHE · 2023-11-01

**Soundness:** 3 good
**Presentation:** 3 good
**Contribution:** 3 good
**Rating:** 8
**Confidence:** 2

**Summary:**

The paper tackles the problem of controlling dynamic systems that involve both continuous and discrete actions (termed "hybrid actions") is challenging in robotics.  The innovative aspect of the authors' solution is the use of the Tensor Train (TT) format, a method to approximate functions with a low-rank tensor. The TT format allows the authors to approximate two crucial components:
State-value function: Represents the value of being in a particular state.
Advantage function: Indicates how much better taking a certain action is compared to others.

**Strengths:**

The paper introduces TTPI, an Approximate Dynamic Programming (ADP) algorithm designed for optimal control. The method leverages Tensor Train (TT) format to address challenges in hybrid system control in robotics.
Traditional control algorithms face problems with systems with non-smooth dynamics and discontinuous reward functions. Existing solutions also often assume differentiability in the system dynamics, which may not always hold.

The paper introduces TTPI, an Approximate Dynamic Programming (ADP) algorithm designed for optimal control. The method leverages the Tensor Train (TT) format to address challenges in hybrid system control in robotics. The experiments show TTPI outperforming other state-of-the-art algorithms in training speed and control performance. A real-world robotics experiment further demonstrates the effectiveness and robustness of this method.

**Weaknesses:**

As the authors themselves mentioned, hand-coding the dynamics in the experiments is very hard to do for complex environments.

TTPI approximates state-value and advantage functions over the entire state-action space, which can result in computational and storage challenges, especially if these functions are not low-rank in the TT representation.

**Questions:**

Tensor operations, especially in high-dimensional spaces, can sometimes introduce numerical instability. How does the TTPI algorithm ensure numerical stability, especially in long-horizon problems?

---

> ### Author Response · Authors · 2023-11-16
> **Response to Reviewer ntHE**
>
> We greatly appreciate your valuable feedback and suggestions. Please find below our response to your question.
>
> > *Tensor operations, especially in high-dimensional spaces, can sometimes introduce numerical instability. How does the TTPI algorithm ensure numerical stability, especially in long-horizon problems?*
> >
>
> In our algorithm, we specify the accuracy of approximation of the value function and advantage function in TT-cross. In our experience, an accuracy of 0.001 is usually enough to avoid any instability. The numerical inaccuracy in the forward simulation is another challenge. In general, TTPI can be robust to this as it is an ADP algorithm (so it approximates the value functions almost everywhere in the state space) and our algorithm does not require long rollouts in the Bellman update (we used rollout of only one step (about 1ms) in the algorithm).
>
> Regarding hand-coding the dynamics, TTPI can work with a simulator as it only requires samples of the tuple (state, action, next-state, reward) queried by TT-cross for modeling the state-value and advantage function. The main requirement from the simulator is that we can query this tuple from an arbitrary state-action pair specified by TT-cross in batch form.  However, as modern simulators such as  NVIDIA Isaac Gym are primarily geared for RL, it still requires some effort in such customization. We are currently investigating this.

---

> > ### Comment · Area_Chair_f8zX · 2023-11-22
> > **Please respond to the author reply**
> >
> > Dear reviewer, please do respond to the author reply and let them know if this has answered your questions/concerns.

---

### Official Review · Reviewer_BNgN · 2023-11-09

**Soundness:** 2 fair
**Presentation:** 3 good
**Contribution:** 3 good
**Rating:** 6
**Confidence:** 3

**Summary:**

This paper develops a method to solve approximate dynamic programming with Tensor Train (TT) representation, which is a compressed tensor to discretely approximate a function. The authors first give a compact and quite neat backgrounds for TT representation and associated operations with TT. The main contribution is a policy iteration algorithm, where the authors replace traditional continuous function approximators (such as using neural network) with TT representations. The authors show the performance of the algorithm using a toy examples in comparison with baseline methods. The method has also been demonstrated in real world robot for a manipulation task.

**Strengths:**

The authors did a good job presenting the necessary background of TT and its associated operations (such as decomposition, rounding, TT-Cross, and TT-go).  I think the authors have fairly discussed the limitation of the method.

**Weaknesses:**

1. My main concern of the paper is the experiment, which is quite limited and there are many remaining questions. Particularly, there are many hyperparameters, it is expected to have ablation study of showing the performance of the method versus the hyperparameters, such as accuracy $\epsilon $ of TT representation, the max TT rank $r_{max}$, discretization resolutions of the state-action spaces....

2. As a reader to implement the algorithm, I wanted to see a straightforward illustration between the running time of the algorithm and dimensions of the state/action spaces of the system, or directly give an overall complexity of the algorithm for one policy iteration.

3. Since the majority of the paper is about the background of TT (previous work), I think the main contribution, which is TTPI algorithm, needs more explanation, including its parallel implementation, which seems a key for applicability of the algorithm. For example, why we need a TT-round operation for the value function in line 8?

**Questions:**

Since in the introduction the authors mentioned RL, I think it would be interesting to discuss the potential of the methods into "model-free" settings.

---

> ### Author Response · Authors · 2023-11-16
> **Response to Reviewer BNgN**
>
> We greatly appreciate your valuable feedback and suggestions. Here, we offer responses to specific questions and concerns that you have thoughtfully presented.
>
> > *My main concern of the paper is the experiment, which is quite limited and there are many remaining questions. Particularly, there are many hyperparameters, it is expected to have ablation study of showing the performance of the method versus the hyperparameters, such as accuracy of TT representation, the max TT rank, discretization resolutions of the state-action spaces....*
> >
>
> > *As a reader to implement the algorithm, I wanted to see a straightforward illustration between the running time of the algorithm and dimensions of the state/action spaces of the system, or directly give an overall complexity of the algorithm for one policy iteration.*
> >
>
> The hyperparameters in TTPI are intuitive and we have observed them to be robust to the changes in the hyperparameters. The four main hyper-parameters are (1) the upper bound on the TT rank $r_{max}$ of the state-value function and the advantage function (2) the accuracy of tt-approximation $\epsilon$ (3) the number of samples used in TTGO in computing the policy (4) the number discretization of the state and action space. We have maintained the same choice of hyperparameters in all the experiments across different robotic tasks, so believe it is quite robust.
>
> In general, due to the properties of TT modeling, the computation involved and memory required grows linearly with respect to the number discretization and the dimensionality of the state-action space, and quadratically w.r.t the rank.
>
> In our experiments, we kept the maximum rank to a large enough value (100 but almost all the experiments never reached the maximum rank in TT-cross modeling over the policy iteration procedure). The accuracy of approximation $\epsilon=0.001$ was sufficient and decreasing epsilon will improve the performance slightly but at a higher computational cost. The number of discretizations of each variable (state or action) is something that depends on the problem at hand and it is fairly straightforward in robotics problems. For example, if it is a joint angle ranging from $(-\pi, \pi)$, a discretization of about 50 to 100 suffices as we employ interpolation for continuous variables. For (3), the higher the number of samples used, the more optimal the policy computation at the cost of linear growth in computational cost and the memory requirement. We set it in the range of 10 to 100.
>
> > *Since the majority of the paper is about the background of TT (previous work), I think the main contribution, which is TTPI algorithm, needs more explanation, including its parallel implementation, which seems a key for applicability of the algorithm. For example, why we need a TT-round operation for the value function in line 8?*
> >
>
> The reason for TT-round operation is that the TT representation obtained by TT-cross need not be optimal in terms of the number of parameters used and the same tensor in TT format can be represented more compactly with the TT-round operation.
>
> We will add more information about the implementation details in the paper.
>
> > *Since in the introduction the authors mentioned RL, I think it would be interesting to discuss the potential of the methods into "model-free" settings.*
> >
>
> We believe our approach can be used for RL and policy learning in general. TTPI can be extended to RL in the following two ways:
>
> (1) Model-free RL:  In this setting, TT as used in TTPI could be used as the function approximator. The difference would be instead of using TT-cross to find the TT representation of the value function or advantage function, we can use gradient-based techniques to find the TT approximation to the value and advantage function (or directly the Q function which will result in Q learning). The policy computation from the advantage model (or Q model) is done as in TTPI. We believe this idea has a huge potential in policy learning in general.
>
> (2) Model-based RL: In this setting, a model of the system is learned using NN or any other function approximator. Then, we can apply the TTPI algorithm using the learned model.

---

> > ### Comment · Reviewer_BNgN · 2023-11-20
> > **Thanks for author's response**
> >
> > I think the authors have positively addressed most of my concerns. I hope those comments could also be reflected in the final draft. I have increased my score from 5 to 6. Thanks.

---

> > > ### Author Response · Authors · 2023-11-23
> > > **Thanks for reviewer's response**
> > >
> > > We thank the reviewer for raising the score! We will include your suggestions in the final draft.

---

### Meta-Review · Area_Chair_f8zX · 2023-12-05

**Metareview:**

(a) The work is dealing with developing approximate DP methods for systems with hybrid action spaces. These involve a mix of continuous and discrete decisions. They show a new type of tensor decomposition method for ADP based on the tensor train method. They validate this on both a simple point domain and a real robot domain.

(b) The strengths of the paper are it's novelty in method and validation on a real robot experiment.

(c) The weaknesses of the paper are the sim experiment being relatively simple, the tensor method potentially requiring a lot of storage/coverage and the somewhat limited description of TTPI.

(d) More complex simulation experiments would certainly help make the paper stronger.

**Justification For Why Not Higher Score:**

The somewhat limited experimentation in simulation, and the writing of the paper not being extremely clear on TTPI detract from it being an oral in my opinion. I think the simulation experiment also could be a lot stronger showing more ablations and comparisons rather than just a single experiment.

**Justification For Why Not Lower Score:**

The three reviewers gave it an 8, and one a 6, ranking it very highly overall. The method is novel, there are real robot results and overall it seems simple and broadly applicable to problems in robotics.

---

### Decision · Program_Chairs · 2024-01-16

Accept (spotlight)